# A Blockchain-Based Trusted Edge Platform in Edge Computing Environment

**DOI:** 10.3390/s21062126

**Published:** 2021-03-18

**Authors:** Jinnan Zhang, Changqi Lu, Gang Cheng, Teng Guo, Jian Kang, Xia Zhang, Xueguang Yuan, Xin Yan

**Affiliations:** Laboratory of Information Photonics and Optical Communications, School of Electronic Engineering, Beijing University of Posts and Telecommunications, Beijing 100876, China; luchangqi@bupt.edu.cn (C.L.); cheng_gang@bupt.edu.cn (G.C.); guotengbupt@163.com (T.G.); jian.kang@bupt.edu.cn (J.K.); xzhang@bupt.edu.cn (X.Z.); yuanxg@bupt.edu.cn (X.Y.); xyan@bupt.edu.cn (X.Y.)

**Keywords:** edge computing, blockchain, microservice, internet of things

## Abstract

Edge computing is a product of the evolution of IoT and the development of cloud computing technology, providing computing, storage, network, and other infrastructure close to users. Compared with the centralized deployment model of traditional cloud computing, edge computing solves the problems of extended communication time and high convergence traffic, providing better support for low latency and high bandwidth services. With the increasing amount of data generated by users and devices in IoT, security and privacy issues in the edge computing environment have become concerns. Blockchain, a security technology developed rapidly in recent years, has been adopted by many industries, such as finance and insurance. With the edge computing capability, deploying blockchain platforms/applications on edge computing platforms can provide security services for network edge environments. Although there are already solutions for integrating edge computing with blockchain in many IoT application scenarios, they slightly lack scalability, portability, and heterogeneous data processing. In this paper, we propose a trusted edge platform to integrate the edge computing framework and blockchain network for building an edge security environment. The proposed platform aims to preserve the data privacy of the edge computing client. The design based on the microservice architecture makes the platform lighter. To improve the portability of the platform, we introduce the Edgex Foundry framework and design an edge application module on the platform to improve the business capability of Edgex. Simultaneously, we designed a series of well-defined security authentication microservices. These microservices use the Hyperledger Fabric blockchain network to build a reliable security mechanism in the edge environment. Finally, we build an edge computing network using different hardware devices and deploy the trusted edge platform on multiple network nodes. The usability of the proposed platform is demonstrated by testing the round-trip time (RTT) of several important workflows. The experimental results demonstrate that the platform can meet the availability requirements in real-world usage scenarios.

## 1. Introduction

Nowadays, the Internet of Things (IoT) is playing a prominent role in the real world by supporting operations and communications autonomously, thereby facilitating and advancing new services that are widely used in everyday life [1]. With the increasing number of access devices, cloud computing technology applied to the Internet of things might inevitably cause long-time latency for users. Meanwhile, a sizeable volume of data produced by heterogeneous IoT devices is transmitted to the cloud for computing and storage service, which require high performance for cloud platforms and a great demand for network bandwidth and have potentially centralized risk [2].

To solve these problems, edge computing is presented. Edge computing is a new paradigm, and it calls for technologies to enable computation to be done at the edge of the network close to data sources [3]. In order to reduce the data volume sent to the cloud and release the pressure of the bandwidth burden, edge computing conducts computing tasks on edge servers to process part of the data. At the same time, it also shortens the response time of the service to the terminal. Edge computing makes a wide range of applications, such as smart city, smart home, intelligent manufacturing, and many other IoT use cases. However, these edge devices may not always cooperate because they are in a distrusted environment [4]. In some cases, malicious edge devices would deny that they have read shared business data from others even though they are benefited from it. Moreover, malicious devices can do the tampering when accessing these business data. These distrust issues eventually cause non-collaboration in the edges. Therefore, the security of edge computing faces a significant challenge [5,6,7].

Blockchain can be defined as a distributed decentralized, shared tamper-resistant database. It is maintained, shared, replicated, and synchronized by multiple participants in a Peer-to-Peer (P2P) network. It can facilitate to establish a secure, trusted, and decentralized intelligent system to solve the security and privacy problems in edge computing [8]. Edge computing can provide computing power and storage space for blockchain.

Therefore, it has become an inevitable trend to integrate edge computing frameworks and blockchain networks in the edge environment [9]. Blockchain and business applications share edge computing node resources, effectively offloading blockchain storage and mining computations to the edge of the network [10]. Furthermore, the off-chain storage and off-chain computation at the edges enable scalable storage and computation on the blockchain [11]. With blockchain services deployed on edge nodes, we can eliminate the information silos between different edges and establish a collaborative mechanism between edge nodes. Meanwhile, blockchain can help establish authentication and access control mechanisms for edge computing systems.

In this paper, we aim to integrate an edge computing framework and blockchain network to provide edge computing clients with a secure authentication mechanism. We implement a trusted edge platform based on microservice architecture, access Edgex Foundry framework to meet the needs of edge computing, and enhance the ability of Edgex through the edge application module. The interaction process with blockchain is abstracted into multiple microservices to form a security authentication module. The security authentication microservices are responsible for authentication and access control. Design the API gateway to parse external requests. Then it routes the request to the endpoints of the internal security authentication microservice and Egdex microservice. The significant contributions of this paper are lists as follows:

The trusted edge platform is seen as an intermediary between the Hyperledger Fabric blockchain network and Edgex Foundry framework. It provides a private network environment that allows for private data exchange among the microservices. The platform supports collaboration among multiple edge computing nodes, and a collaborative process is designed for this purpose.

The trusted edge platform is organized in a microservice architecture with excellent scalability. The platform uses Docker containers to build services. The introduction of the EdgeX Foundry framework enhances the ability of the platform to handle heterogeneous data.

The proposed platform access to the Hyperledger Fabric blockchain network introduces reliable security mechanisms for the edge environment. The identity information of both the terminal and the user is stored on the blockchain. External requests need to be authenticated and access controlled before they can be processed by internal services.

The rest of the paper is structured as follows. Section 2 explores related works in the use of blockchain technology in IoT platforms. Section 3 illustrates the details of the proposed trusted edge platform. Section 4 discusses the workflow of typical functions in the trusted edge platform. Section 5 reports an extensive experimental study using test scenarios that are built on both Raspberry Pi and Desktop. The conclusions are drawn in Section 6.

## 2. Related Works

In this section, we will survey the latest frameworks of integrated blockchain and edge computing Systems. The authors of [12] propose a distributed blockchain cloud architecture with edge computing (fog nodes at the edge of the network) enabled by Software-Defined Networking (SDN), which is categorized into three layers, i.e., device, fog, and cloud. All the SDN controllers are connected using blockchain technology in a distributed manner. If required, a fog node reports the outcomes of processing data to the distributed cloud and the system layers, accesses the cloud to deploy the application service, and when there are not enough computing resources, discharges computing workloads to the cloud. This architecture solves many of the problems of traditional cloud computing, such as time-to-data transfer, scalability, security, and high availability.

In [13], a blockchain-based multi-layer IoT network model is proposed. The whole IoT model consists of two parts: edge layers and high-level layers. The edge layer is defined as a local area network consisting of a certain number of objects with a central node managing them. There is no central node at the high level, so each node is data independent. The management of data within each node is carried out by itself. However, the data exchange between nodes is recorded by a blockchain distributed ledger maintained by all nodes. The multi-layer network model based on blockchain technology provides a feasible solution for the establishment of a wide-area secure IoT network.

The authors proposed a video surveillance system in [14] that uses blockchains, edge computing, InterPlanetary File System (IPFS) technology, and convolution neural networks. Blockchain technologies enhance the reliability and robustness of the system. Edge computing is used to collect information and process and interpret sensory data from large-scale wireless sensors. In order to achieve large video data storage and real-time monitoring, both IPFS and CNNs are used.

In order to solve the problem of edge computing security in IIoT, the authors of [15] develop a mechanism focused on blockchain identity management and access control. The registration and authentication of network entities are implemented using self-certified password technology. They suggested a protocol for a light-weight key agreement based on self-certified public keys that provide IIoT authentication, auditability, and confidentiality. 

In [16], the authors used mobile communications, computing, and caching (3C) systems to improve the bandwidth and latency of wireless communications. They note that smart city applications have enormous data transmission needs, which can cause significant transmission bandwidth pressure. Unless edge computing and caching technologies are used, the current wireless infrastructure will be quickly depleted. Then they developed a blockchain database to solve the security problem of communication between smart cities and home devices and sensors. 

In [17] and [18], the authors proposed a blockchain architecture based on IoT virtual resources on the edge host. It allows the fog node as an extension of cloud computing. Further, the authors pay attention to how the local edge network configures its P2P-communicated M2M applications. Virtualization of software-defined IoT components (virtual resources) is implemented to handle the configuration of a large and heterogeneous collection of devices. The configuration of virtual resources (code or metadata) needs to be encrypted and stored on the blockchain. Meanwhile, multiple tenants registered in the permission-based blockchain can identify and deploy their virtual systems and read or write in blocks. Therefore, the approved blockchain manages the provisioning of virtual resources and multi-tenant access in a secure manner.

## 3. Trusted Edge Platform Design Based on Blockchain in Edge Computing

Based on the design idea of microservice architecture, we propose a trusted edge platform based on a blockchain network in an edge computing environment. As shown in Figure 1, the edge computing environment consists of the following participants:

The terminal is a device that connects the physical world and cyberspace. They can be classified into smart devices and resource-constrained devices based on their computing power. Smart devices are capable of performing various functions such as initial data processing, encryption, and transmission, while resource-constrained devices are only capable of sensing and collecting environmental data. In the edge computing environment, we abstract the terminals into two types of devices: acquisition devices and execution devices. The terminals are connected to the proximate edge computing servers through multiple communication protocols, such as NB-IoT, LoRa, WiFi, and NFC. The terminal senses the changes in the environment through the RF module or sensor module at the front end, and after calculation, makes decisions on the required response.

Edgex Foundry, an open-source edge computing framework, connects to various IoT end devices through different protocols, manages them and collects data from them, and exports the data to applications at the edge or in the cloud for further processing [19,20,21,22]. edgex consists of a series of logically independent microservices that are divided into four service layers and two base enhancement systems services. The four service layers include Device Service Layer, Core Service Layer, Support Service Layer, and Export Service Layer. The two base enhancement system services are Security and System Administration. Each layer contains multiple components that communicate with each other through Restful API interfaces. In the edge computing environment, we extract the critical functions required for the conceptual definition from Edgex and combine them into the proposed platform. Then, from the user’s perspective, the functionality of the edge computing framework can be abstracted as a collection of APIs.

The blockchain networks build secure, trusted, decentralized intelligent systems in an edge environment to solve security and privacy problems in edge computing. Multiple edge nodes can construct a blockchain network. Part of the terminal devices is limited to their computing power. Thus, they are not parts of a blockchain network. The identity information of all these terminals is stored in the blockchain network. After the blockchain verifies the identity information, authorized terminal entities can read the blockchain data but can only write by the edge server. It contributes to reducing the excessive exposure of data. The edge servers can edit smart contracts to create access control policies specific to their needs. In the edge computing environment, device identity information, access control information, and access control policy are stored by a decentralized blockchain network, which can effectively prevent data from being tampered with.

Hyperledger Fabric is a federated chain platform that allows multiple parties to participate, develop, deploy and run blockchain applications [23]. Hyperledger Fabric aims to create a modular and scalable blockchain development framework that provides solutions for enterprise-class blockchain applications. The Hyperledger Fabric blockchain system consists of the following components: Client, CA, Peer, and Orderer. The client is the access point between users and the Hyperledger Fabric network, on which a proprietary SDK is deployed, and users can use the client to initiate transaction requests. The ca node is the certificate authority center of Hyperledger Fabric, responsible for adding users, issuing registration certificates, and renewing and revoking certificates. Peer is the main body of ledger maintenance in the Hyperledger Fabric network, and some Peers act as Endorser to endorse the legitimacy of the transaction content [24]. The order node receives all transactions from the network and sorts and packages them into blocks according to time [25]. Order is not involved in the execution and validation of transactions and therefore does not care about the specific content of transactions. The goal of the order is to agree on the order in which transactions are generated and broadcast this result. The layered trust model consisting of Endorser and Order decouples the consensus process into two parts: the application layer trust and the underlying consensus. The architectural advantages of HyperLedger Fabric enable easy deployment on edge servers. It is also easier to organize collaboration between multiple nodes using Hyperledger Fabric.

In order to integrate the capabilities of the edge computing framework and the blockchain network more efficiently, we propose a trusted edge platform based on microservice architecture. This platform has the characteristics of light-weight, ease of deployment, and high security. The proposed platform consists of three types of microservices, as illustrated in Figure 2. In the edge computing environment, the Edge Application Module acts as an access object for the rule engine of Edgex Foundry to provide extended services. Based on the Hyperledger Fabric blockchain, the Edge Application Module is able to build collaborative workflows among multiple edge servers to enable cross-domain cooperation. Security authentication implements registration services, identity authentication, security management, and access control with the help of the Hyperledger Fabric blockchain network. API gateway orchestrates services within the platform and exposes them as network interfaces for terminals and users to invoke.

### 3.1. Edge Application Module

The edge application extends edge-side business capabilities to close the loop on local data by combining with the Edgex Foundry framework. The edge application module allows users to define operational functions to meet terminal requirements and run complex models such as neural networks. Edge computing resources are fully dispatched through the collaboration of several microservices as described below:

(1) Application Process: The Application Process Service responds to requests for tasks from terminals. The service consists of the following three components: Task Scheduler, Task Queue, and Task Manager. The task scheduler accepts task requests and generates an ID for each task to be executed. The ID is used as a key to store the task name and source data in the database. Thereafter, the ID is inserted into the task queue by the scheduler. The scheduler can also edit the scheduling rules to execute some specific tasks at regular intervals. The task queue manages the ID of all pending tasks, and the data in the task queue can only be accessed by the task scheduler and the task manager. The task manager gets the task ID from the task queue, retrieves the task information in the database based on the ID, and executes them. The manager has an overload handling policy, a thread selection policy and, an idle policy to control the consumption of platform resources by tasks.

(2) Task Registration: Task registration service provides a task registration process for terminals and platforms. The registration service first calls the authentication service in the security authentication module to distinguish the service level. The registration service maintains a task registration table that records information about each user’s registered tasks. The collaborative process service can access this table to get information about the required tasks for building collaborative processes.

(3) Collaborative Process: This service is used to organize multiple edge computing nodes to achieve collaborative work. The Edgex Foundry platform provides good interoperability locally, but for multi-node collaboration, it requires the participation of a third-party platform to organize the services. The collaboration process service can be free from the third party, and several edge nodes can organize the service by themselves. Multiple edge nodes elect a leader node through the Raft algorithm. The leader node has the control authority of the whole collaborative process and can create Hyperledger Fabric channels and edit smart contract templates. The leader node integrates the edits made to the smart contract by the member nodes and verifies their validity. After the leader node deploys the smart contract, the nodes in the channel work according to the constraints of the smart contract, enabling collaboration among multiple nodes.

(4) Database: Database is used to store applications and provides caching services for them during their work. The database has access rights and opens up different add, delete, and check functions to users with different identities.

### 3.2. Security Authentication Module

The security authentication decoupled the security functions into multiple microservices and deployed them on Docker containers. These decentralized security microservices work as a service cluster to offer a scalable, flexible, and light-weight data sharing and access control mechanism for the edge computing system. The key service components and operations are introduced below:

(1) Registration Service: In the proposed edge platform, before accessing Edgex Foundry services, all terminal entities must submit a registration request to the registration microservice. The registration service performs the SHA-256 algorithm to generate InfoHash, a hash of identity information used as authentication credentials when authenticating. The Hyperledger Fabric blockchain creates a new peer node. It binds the peer node to the terminal, which means that the peer and the terminal share the public-private key pair. peer node invokes a smart contract to save the InfoHash in the blockchain and generates and returns the VID which uniquely identifies the terminal’s identity data. The terminal stores the public-private key pair and the VID locally.

(2) Identity Authentication: The terminal submits the identity data identifier (VID), the identity plaintext data (Info), and the digital signature of the private key (sign (sk, Info)) to the identity authentication service. The identity authentication service obtains the corresponding InfoHash data through the smart contract based on the VID. The identity authentication service calculates whether the Hash (Info) is equal to the InfoHash to ensure that the Info provided by the terminal is consistent and complete with the stored identity data. The identity authentication service ensures that the terminal has the corresponding private key by verifying the digital signature of the terminal. After authenticating the identity data of thr terminal, the authentication service invokes the smart contract to record the authentication process. After completing the above work, the authentication service returns the authentication result.

(3) Access Control: In order to successfully access a service or resource at a service provider on a trusted edge platform, the terminal needs to send an access permission request to the access control service to obtain a permission token. The access control service runs an authorization policy to evaluate the access request based on the identity information of visitor. If the access request is authorized, the access control service invokes a smart contract to update the evaluation result on the blockchain and returns a querying token. When the service provider receives a request from an endpoint, the service provider determines whether to accept the service by whether the request carries a token. After that, the authorization data on the blockchain is queried by the token to determine whether the service is provided to the terminal.

(4) Security Management: In the Security Authentication module, the Security Management microservice acts as a data and security service manager, managing the registered data and access control policies. The policies for access control are maintained and updated by the security management service. When a device is registered, the security management service sets responsive permissions for each endpoint based on a predefined permission policy. The edge server administrator can modify the information of registered devices through the security management service, while ordinary users can only modify the information of their own registered terminals. The accessed privileges of the terminal can be modified by the administrator and its own registered users, while the access privileges can only be modified by the administrator.

### 3.3. API Gateway

The API gateway extends the device access layer of the Edgex Foundry framework to add security control mechanisms while meeting terminal access needs. The API gateway provides an HTTP listener to receiving requests from clients. There are several essential components, such as the controller, which is used by the API gateway to parse the requests it receives; the handler, which provides business logic by calling the internal microservices of the platform; the configuration, where the configuration information of APIs is stored; and logs, which records the key historical data of API gateway.

## 4. Key Process Design of Trusted Edge Platform

The key to the proposed platform design is how to take advantage of the edge computing platform and blockchain. Edgex Foundry has strong interoperability and is responsible for data collection, integration, and heterogeneous processing in the whole system. Hyperledger Fabric deployed using microservices architecture has better compatibility with the proposed platform. The following will elaborate on how the platform integrates edge computing and blockchain.

After system initialization, the user initiates a terminal device registration request to the API gateway. The request is forwarded to the registration microservice through the processor of the API gateway. The registration microservice hashes the terminal identity information transmitted by the user and obtains a virtual ID (VID) as proof of identity authentication. The registration service calls the Hyperledger Fabric service to create peer nodes and extract their public-private key pairs. When the peer node completes smart contract deployment, the registration service initiates a transaction process to store the InfoHash into the blockchain. The virtual ID (VID) used to query the InfoHash is returned when the transaction is completed. The registration service returns the public-private key pair and VID to the terminals. After that, the handler invokes the device service API of Edgex to register the necessary information and data communication configuration of the terminal. Figure 3 shows the sequence diagram for device registration.

When accessing the authentication microservice, the terminal needs to submit the identity data identifier (VID), identity data (Info), and the terminal private key digital signature (sign (sk, Info)). The identity service obtains the corresponding InfoHash identity data through a smart contract based on the VID. After hashing Info and comparing whether Hash (Info) is equal to InfoHash, the authentication service can determine that the Info provided by the terminal is consistent and complete with the stored identity data. The identity authentication service ensures that the terminal has the corresponding private key by verifying the digital signature and proves that the identity data belongs to the terminal. After completing the above operation, the authentication service invokes a smart contract to record this authentication service on the blockchain. Finally, the authentication service returns the authentication result. Figure 4 illustrates the sequence diagram for identity authentication.

The terminal needs to determine its access rights before requesting services or obtaining resources. The API Gateway detects whether the endpoint request contains a token, and if there is no token, then the access control microservice needs to be invoked. The API gateway sends the terminal information and the information of the accessed service to the Access Control microservice. First, the access control service obtains the latest authorization policy from the security management microservice. After that, it runs the authorization policy and evaluates whether the visitor meets the authorization conditions. If the assessment is passed, the access control service invokes a smart contract to update the assessment result to the blockchain. After the smart contract is executed, it returns a token used to query the verification result. Finally, the access control microservice returns the token data. Figure 5 illustrates the sequence diagram for access request.

Before the terminal officially works after registration, users need to deploy application services on the Edge Application module. The user accesses the interface exposed by the API gateway and transmits the information about the target task. The Handler component simply processes the task information and forwards data to the task scheduler. The scheduler temporarily stores the task data in the database and generates a unique ID to identify the task. The task scheduler inserts the task ID into the end of the task queue. The task manager takes out the task ID from the task queue, then extracts the task information from the database and loads the task resources. After waiting for a while, the task starts to execute, and Edgex Foundry will export the data to the task data interface. Figure 6 illustrates the sequence diagram for smart application binding.

In order to increase the interaction between edge computing nodes, the Trusted Edge Platform provides collaborative process microservices to help multiple nodes establish cooperative relationships. The Channel mechanism of Hyperledger Fabric has data isolation, and multiple nodes can join the same Channel. The cluster of nodes elects the leader node based on the draft algorithm. After the leader node of the cluster is elected, the leader is responsible for organizing the members to write, verify and deploy the smart contract. The cluster then works according to the smart contract. Figure 7 illustrates the sequence diagram for the collaborative process.

## 5. Experimental Study

We build the edge computing network with two different performance hardware devices. A desktop with a 2.3 GHz Intel Core i7 (8-core) CPU and 16 GB RAM memory capacity meets our needs due to the power of the edge server. The low power consumption and the compact size and portability make Raspberry Pi a prime candidate for edge processing. We deployed our edge computing environment using a Raspberry Pi 4 with a 1.7 GHz Cortex-A72 CPU and 4 GB of RAM. On the software side, we used Docker and Docker Compose to simplify the operation of the application. The details of the hardware and software specifications are presented in Table 1.

To evaluate the usability of the proposed platform, a physical network environment is carried out on a physical network environment, which includes three desktops and two Raspberry Pis. Each device is deployed with an edge computing and blockchain environment. We use IoT end devices simulated by node-red. Edgex Foundry is deployed based on the Geneva version. Hyperledger Fabric version number is 2.1.0. Hyperledger Fabric blockchain network consists of five Fabric organizations. There are seven Orders randomly deployed on five physical devices in the whole network. Each organization initially has only one peer node. We use Docker Swarm to provide an overlay network layer for containers across hosts to enable containers to communicate with each other. A Schematic diagram of the experimental equipment organization scheme is shown in Figure 8.

To measure the general cost incurred by the proposed platform in terms of edge device processing time, we selected two desktops and a Raspberry Pi for a multi-node interoperability test. The Raspberry Pi acts as the data generator, capturing images through a RaspiCam camera connected to the Edgex Foundry platform. One desktop acts as the data processing side, invoking the image recognition model to judge the images. On the command execution side, the other desktop responds to the image recognition result and sends the command to the simulation terminal through Edgex Foundry. The experimental design is shown in Figure 9. To test the platform performance, we rent the cloud servers from the cloud service provider and deploy the image recognition service. Besides, we conduct tests based on the key processes proposed in Chapter 4. We performed 50 test runs of the designed test scenario and recorded its time consumption and resource usage.

### Result Analysis

To evaluate the usability of the designed platform in a real-world environment, we performed Round-Trip Time (RTT) tests on the system workflows in Section 4. We compared the measured data with human reaction times [26,27], which measure the reaction time of a person to a certain stimulus. If the experimentally measured RTT is less than the reaction time, it indicates that the proposed platform is applicable in the real world. The statistical average reaction time obtained from tens of millions of records is 215 milliseconds.

Comparative experiments on multi-node collaboration tasks are conducted in a stable network environment. The experiments measured the time required to complete the task flow at different acquisition frequencies. The acquisition frequency of the camera gradually increases from one frame per second to 10 frames per second. The usability of the proposed platform when collaborating among multiple nodes is judged by comparing the two sets of test data. The experimental data statistics are shown below.

The author of [28] proposes a microservices secure agent platform that integrates edge computing platforms with API gateway technology. It verifies the availability of the system by measuring the RTT of the REST API interface. The average elapsed time for all system functions is between 3 ms and 51.7 ms, which meets the practical availability requirements. As can be seen in Figure 10, the time consumed by the platform designed in this paper is between 25 ms and 183 ms. Considering the time consumption of the blockchain network, the time consumption of the proposed platform in this paper is within the manageable range. Also, due to blockchain technology, this paper has more advantages in secure storage, identity authentication, and interoperability of edge nodes. It indicates that our solution is better than the existing solutions.

The data in Figure 11 shows that when the camera is located at a low sampling rate, the task flow done by edge computing based collaboration is about the same in terms of time consumption as the task flow done by cloud based services. The blockchain-based security mechanism running at the edge side consumes some of the time. However, when the camera acquisition frequency increases, the service latency of cloud services rises rapidly, while the rise of edge computing is relatively slow. It can be seen that the use of edge computing collaboration can increase the service response speed with greater throughput.

## 6. Conclusions

In this paper, we propose a trusted edge platform for integrating edge computing and blockchain, which uses a microservice architecture to address privacy security challenges in edge environments through blockchain-based authentication and access control mechanisms. A conceptual validation prototype is constructed in a physical edge computing network environment for verifying the feasibility of the proposed platform. In the edge environment where terminal, edge computing, and blockchain participate simultaneously, the trusted edge platform acts as a hub to connect three parties. The API gateway handles external access requests and routes them to internal microservices. The security authentication module is connected to the Hyperledger Fabric blockchain network and is responsible for user/terminal identity authentication and access control. The edge application module is used to improve the intelligence of the Edgex Foundry framework on which different types of applications can be deployed. With the features of Hyperledger Fabric blockchain, we designed a multi-node inter-collaboration process on the platform. Extensive experimental studies have been conducted with satisfactory results. It verifies that the trusted edge platform can effectively authenticate and control access policies in real-time in the edge environment. The intelligent applications on the platform can promptly respond to requests from terminals. This work shows that the proposed trusted edge platform is a promising approach. It can provide portable, scalable, and fine-grained security mechanisms for integrating edge computing and blockchain. It can also improve the coverage of blockchain networks in edge computing environments.

While the reported work meets practical applications, there is still a long way to provide a complete decentralized and light-weight security solution to integrate edge computing and blockchain. Our subsequent work aims to focus on the research of light-weight consensus algorithms for blockchain.

## Figures and Tables

**Figure 1 sensors-21-02126-f001:**
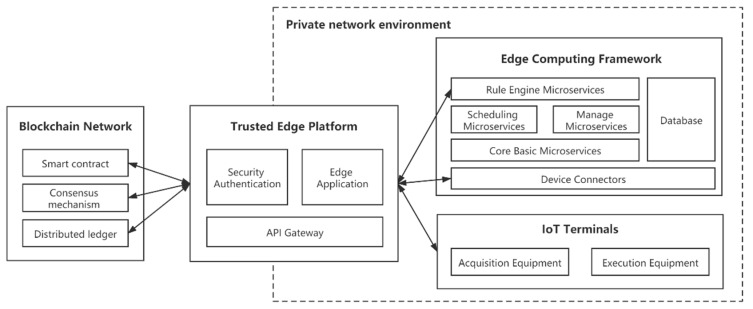
Edge Computing Environment Configuration.

**Figure 2 sensors-21-02126-f002:**
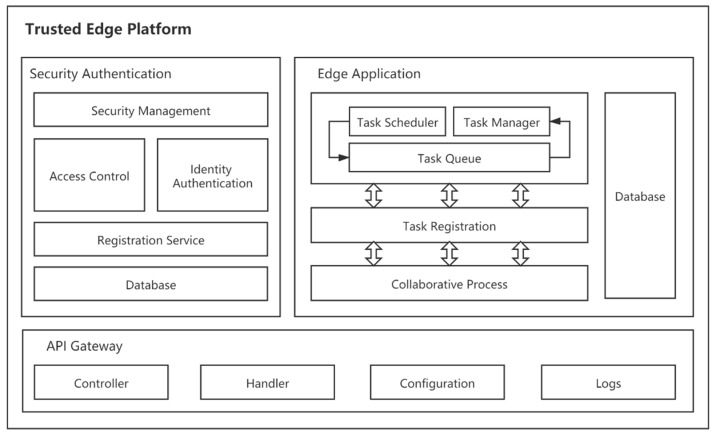
The architecture of the trusted edge platform.

**Figure 3 sensors-21-02126-f003:**
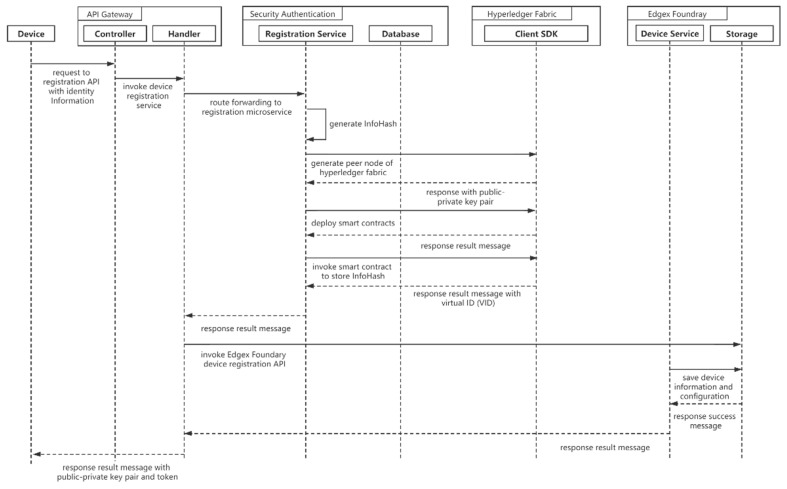
Sequence diagram for device registration.

**Figure 4 sensors-21-02126-f004:**
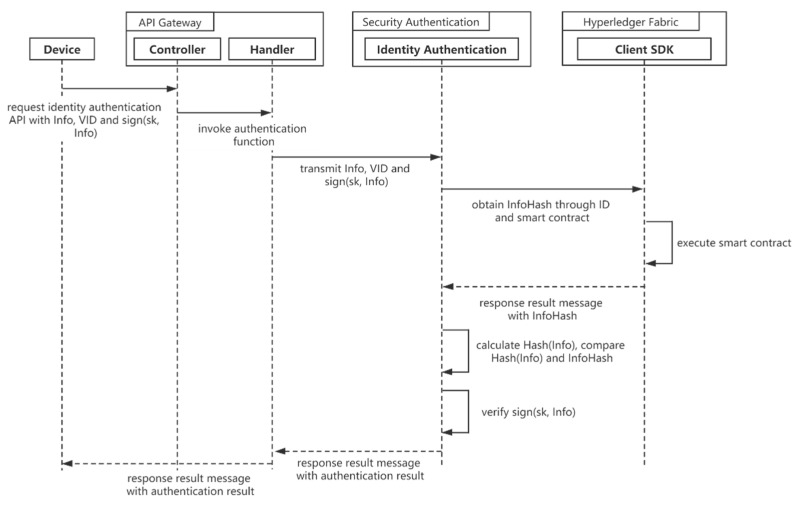
Sequence diagram for identity authentication.

**Figure 5 sensors-21-02126-f005:**
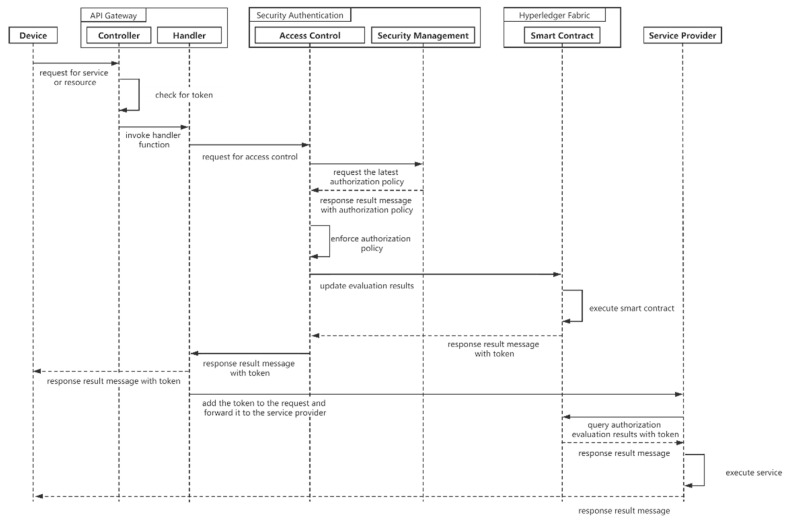
Sequence diagram for access request.

**Figure 6 sensors-21-02126-f006:**
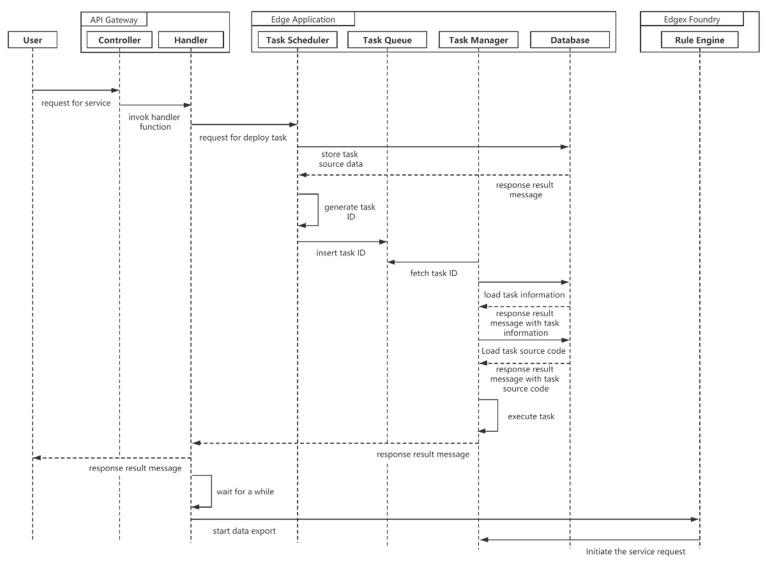
Sequence diagram for smart application binding.

**Figure 7 sensors-21-02126-f007:**
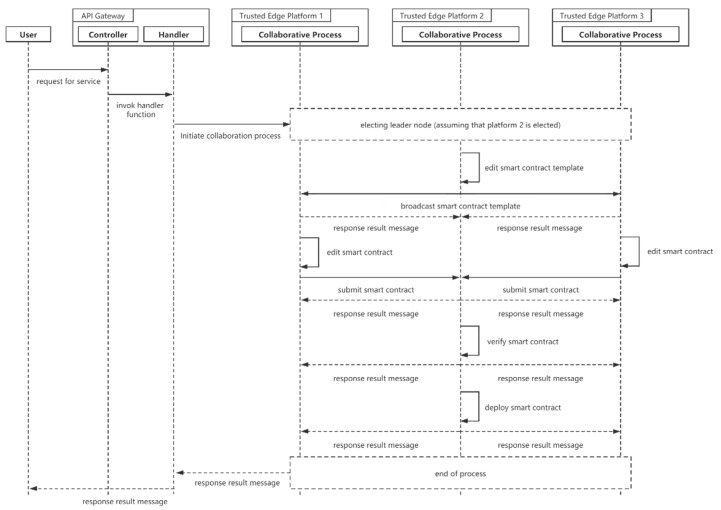
Sequence diagram for the collaborative process.

**Figure 8 sensors-21-02126-f008:**
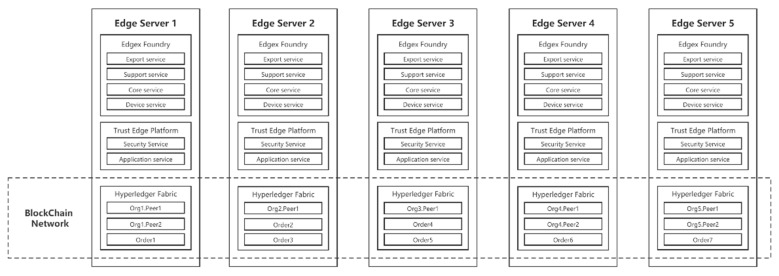
Schematic diagram of experimental equipment organization scheme.

**Figure 9 sensors-21-02126-f009:**
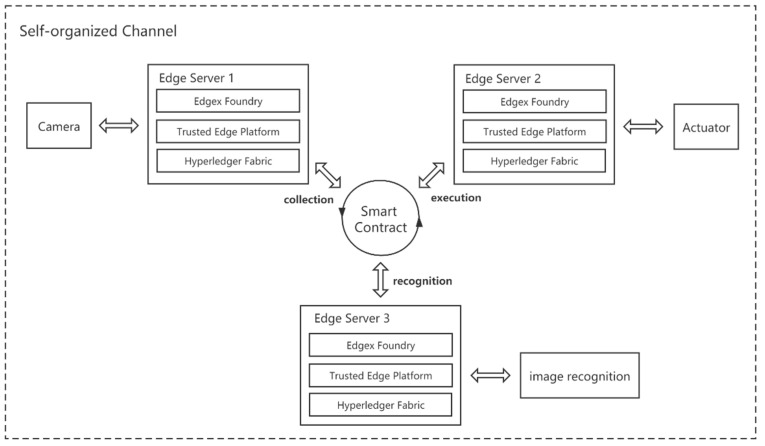
Schematic diagram of collaboration experiment between multiple nodes.

**Figure 10 sensors-21-02126-f010:**
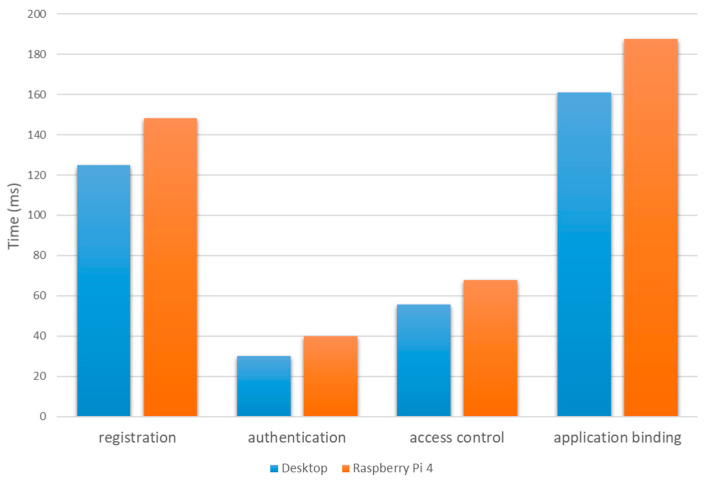
Statistics of average time consumption of platform key processes.

**Figure 11 sensors-21-02126-f011:**
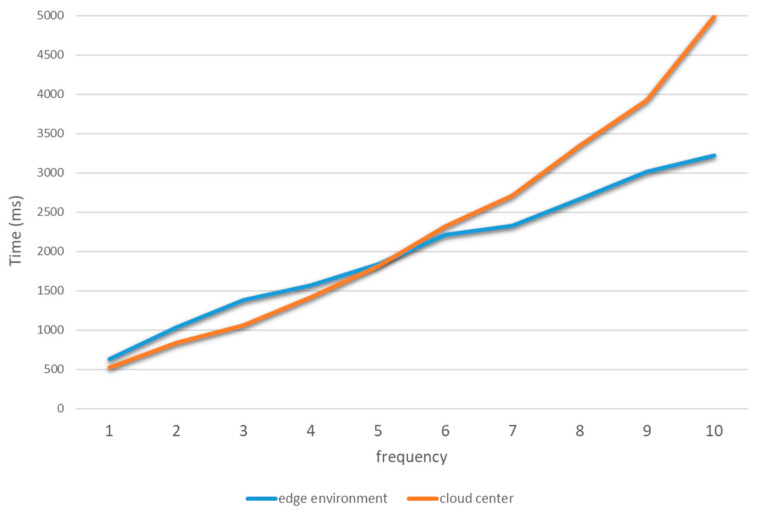
Average time consumption of multi-node cooperative experiment.

**Table 1 sensors-21-02126-t001:** Hardware and software specififications for edge computing network architecture.

Item	Specification	Description
Desktop	OS	Ubuntu 18.04
	CPU	Intel Core i7
	Memory	4 GB
	Hard Disk	1 TB
Raspberry Pi 4	OS	Ubuntu 18.04
	CPU	BCM2711
	Memory	4 GB
	Hard Disk	128 GB
Software	Library	Golang, JavaScript
	Application	Docker, Docker Compose

## Data Availability

The raw/processed data required to reproduce these findings cannot beshared at this time as the data also forms part of an ongoing study.

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
