# Peer review of "A Blockchain-Based Trusted Edge Platform in Edge Computing Environment"

_sensors, 2021, doi:10.3390/s21062126_

Round 1

Reviewer 1 Report

The concept is good and relevant timely, similar to what is shown at “Sensors 202121(5), 1559; https://doi.org/10.3390/s21051559” . However, the security you talk about blockchain can be performed with a simple token exchange like in OAuth 2.0 and SSH much lighter and secure as already proven. For example, the same concept which is presented about trusted and distribute data can simply be implemented with a YAML file and SSH tokens executed under open-source tool Ansible for distributed architecture, but faster and simpler compared to proposed architecture due to eliminating extra layers of microservices like dockers and Kubernetes.

Furthermore, the authors demonstrate the concept only between a pc and raspberry, the question is: where is the distributed cluster architecture?

The authors should reconsider the proposed solution and test it at the list in a distributed cluster of edge computers before publishing

Author Response

Dear Editor,

Thank you for your kind letter of ' A Trusted Edge Platform Based On Microservice Architecture In Edge Computing' on February 25, 2021. We revised the manuscript in accordance with the reviewers' comments and carefully proof-read the manuscript to minimize typographical, grammatical, and bibliographical errors.The attachment is our explanation of the revision based on the reviewer’s comments.

Thank you and all the reviewers for the kind advice.

Sincerely yours,
Changqi, Lu

Reviewer 2 Report

This work aims to integrate an edge computing framework and blockchain network to provide edge computing clients with a secure authentication mechanism. - The combination of edge and blockchain is not new, there are something new are expected for this. Also, the keyword - blockchain needs to be mentioned in the title. - In the evaluation, it is unclear how to implement the blockchain. - A comparison with similar studies need to be considered. - Writing needs to be improved under a professional editing

Author Response

Dear Editor,

Thank you for your kind letter of ' A Trusted Edge Platform Based On Microservice Architecture In Edge Computing' on March 1, 2021. We revised the manuscript in accordance with the reviewers' comments and carefully proof-read the manuscript to minimize typographical, grammatical, and bibliographical errors.The attachment is our explanation of the revision based on the reviewer’s comments.

Thank you and all the reviewers for the kind advice.

Sincerely yours,
Changqi, Lu

Round 2

Reviewer 1 Report

The paper significantly improved after the text additions and the visual explanatory diagrams, therefore I believe it is suitable for further consideration to be published in sensors journal

Reviewer 2 Report

no more comments